# Comparative Transcriptome Analysis Provides Insights into the Effect of Epicuticular Wax Accumulation on Salt Stress in Coconuts

**DOI:** 10.3390/plants13010141

**Published:** 2024-01-04

**Authors:** Xiwei Sun, Ghulam Abid Kaleri, Zhihua Mu, Yalan Feng, Zhuang Yang, Yazhu Zhong, Yajing Dou, Hang Xu, Junjie Zhou, Jie Luo, Yong Xiao

**Affiliations:** 1Coconut Research Institute, Chinese Academy of Tropical Agriculture Sciences, Wenchang 571300, China; sunxw065@163.com (X.S.); fengyalan0707@126.com (Y.F.); zhongyazhua@163.com (Y.Z.); douyj2018@163.com (Y.D.); 2College of Breeding and Multiplication, Hainan University, Sanya 572025, China; abidalik93@gmail.com (G.A.K.); mbj937@hotmail.com (Z.M.); sand_zhou@hainanu.edu.cn (J.Z.)

**Keywords:** coconut, transcriptome, salt tolerance

## Abstract

The coconut is an important tropical economical crop and exhibits high tolerance to various types of salinity stress. However, little is known about the molecular mechanism underlying its salt tolerance. In this study, RNA-Seq was applied to examine the different genes expressed in four coconut varieties when exposed to a salt environment, resulting in the generation of data for 48 transcriptomes. Comparative transcriptome analysis showed that some genes involved in cutin and wax biosynthesis were significantly upregulated in salt treatment compared to the control, including CYP86A4, HTH, CER1, CER2, CER3, DCR, GPAT4, LTP3, LTP4, and LTP5. In particular, the expression of CER2 was induced more than sixfold, with an RPKM value of up to 205 ten days after salt treatment in Hainan Tall coconut, demonstrating superior capacity in salt tolerance compared to dwarf coconut varieties. However, for yellow dwarf and red dwarf coconut varieties, the expression level of the CER2 gene was low at four different time points after exposure to salt treatment, suggesting that this gene may contribute to the divergence in salt tolerance between tall and dwarf coconut varieties. Cytological evidence showed a higher abundance of cuticle accumulation in tall coconut and severe damage to cuticular wax in dwarf coconut.

## 1. Introduction

The coconut (*Cocos nucifera* L., 2n = 32) holds significant value as a tropical crop within the Arecaceae family [1]. It is well known as “the tree of life” due to its numerous uses for economic purposes [2]. Every part of the coconut palm carries value, including its nutritious water [3], mesocap as fiber, trunk as building materials, endocarp as charcoal [4], and endosperm for oil [5]. Coconuts are also used in medicine [6], manufacturing [7], and biofuel production [8]. Cultivated in 93 countries [9], coconut palms can be found along coastlines in tropical regions such as Central and South America, East and West Africa, Southeast Asia, and the Pacific Islands, spanning a total cultivation area exceeding 12 million hectares [10]. As a salt-tolerant crop found globally, soil chloride deficiency can negatively affect the growth of coconut. Soil chloride deficiency usually occurs in rainforest and coastal areas worldwide [11]. A certain concentration of salt (approximately 3.8 kg/tree/year) was used as a fertilizer in chloride-deficient soils to enhance coconut fruit yields in Davao Mindan (Philippines) [12]. However, as a common phenomenon that can be observed worldwide, salinity usually occurs in arid and coastal areas [13] and can come from varies of sources such as soil, irrigation, seawater, or an aquatic environment [14]. Salinity poses a significant challenge to contemporary agriculture, hindering and adversely affecting the growth and progression of crops [15]. As a plant with a strong salt tolerance, coconut accumulates osmotic substances in response to salt stress, leading to a reduction in osmotic potential and water loss [16]. The Hainan Tall coconut, an indigenous variety found on Hainan Island, demonstrated superior capacity for salt tolerance compared to dwarf coconut varieties such as aromatic coconut, yellow dwarf, and red dwarf [17]. The classification of all dwarf plants, symbolizing the complete geographical spread of the crop, alongside tall plants from Southeast Asia and the Pacific, and the distribution of alleles between tall and dwarf varieties, imply that dwarf plants originated from tall forms, specifically those from Southeast Asia and the Pacific Islands [18]. Investigations into potential molecular mechanisms that contribute to salinity tolerance in coconuts are valuable as this knowledge could contribute to the genetic breeding of salt-tolerant coconut varieties.

High soil salinity, a significant environmental stressor, has demonstrated negative effects on plant growth and crop yield. Currently, on an annual basis, 1.5 million hectares of land become unsuitable for agricultural production due to high salinity. This is due to both natural factors such as seawater backflow and human factors such as excessive use of groundwater for irrigation. There are also natural factors such as the inundation of coastal land by tidal water and human factors such as irrigation with salt-containing water [19]. Salinity stress triggers various physiological, biochemical, and molecular changes in plants [20]. These include the regulation of salt uptake by roots [21], the increased accumulation of osmolytes [22], the modulation of reactive oxygen species (ROS) levels [23], the thickening of the cuticular wax layer [24], and modifications in gene and transcription factor (TF) expression associated with salt stress [25]. The cuticle, the lipid layer on the plant’s epidermis, plays a pivotal role in water retention and consists of waxes and cutins. These substances are derived from long-chain fatty acids, predominantly of the C16 and C18 varieties [26].

Research findings indicate that NaCl exposure significantly increases the lipid content of the cuticle in *Arabidopsis thaliana*, which coincides with the accumulation of transcripts for the wax biosynthesis gene ECERIFERUM 1 (CER1) [27]. The CER1 protein is part of an integral cell membrane aldehyde decarbonylase. It is also the core component in long-chain alkane synthesis complexes and participates in the biosynthesis of epidermal wax [28,29]. Similarly, an increased cuticle thickness has been found in *Camellia sinensis* [L.], O. Kuntze [30] and cucumber [31] under salt stress. Therefore, the thickening of the cuticle appears to be a universal adaptive response for plants under salt stress.

Previous studies have been conducted to identify genes associated with wax biosynthesis pathways, including the CER6/CUT1 gene encoding 3-ketoacyl-CoA synthetase [32], the CER8/LASC1 gene encoding long chain acyl-CoA synthetase [33], the GL8 gene encoding very-long-chain 3-oxoacyl-CoA reductase [34], and the wax transporter CER5/ABCG12 [35], as well as wax biosynthesis transcription activating factors MYB30 [36] and MYB96 [37]. These studies have revealed that abiotic stress significantly affects wax biosynthesis and composition in plants. The overexpression of WXP1 [38] and CER1 [39] have been shown to induce the thickening of cuticular wax, subsequently enhancing resistance to abiotic stress by mitigating water loss and reducing cuticle permeability. Furthermore, a notable increase in the accumulation of thicker cuticular wax has been observed in peanuts [40] and jojoba [41] under salt stress. Additionally, overexpressing SHN1/WIN1 has been demonstrated to elevate wax production and accumulation in response to drought stress. In conclusion, cuticular wax plays a crucial role in plants’ responses to abiotic stress [42].

In this study, RNA-Seq was employed to investigate the response of different coconut varieties under salt stress in terms of gene expression. Comparative transcriptome analysis revealed that the deposition of cuticular wax may be a crucial factor in explaining the difference in salt tolerance between tall and dwarf coconut varieties. Moreover, cytological analysis indicated that the accumulation of cuticular wax varied across different coconut varieties following salt treatment.

## 2. Results

### 2.1. Physiological Changes in Four Coconut Cultivars under Salt Stress

Four coconut cultivars, specifically Hainan Tall, yellow dwarf, red dwarf, and aromatic coconut, were exposed to a 300 mmol/L NaCl solution. Two dwarf coconut cultivars manifested clear signs of salt damage, specifically the yellowing and desiccation in mature foliage. In contrast, Hainan Tall exhibited high salt resistance showing no visible signs of salt stress (Figure 1A). To gain deeper insights into the physiological changes under salt stress, soluble sugars, superoxide dismutase (SOD), Malondialdehyde (MDA), and H_2_O_2_ were measured in both Hainan Tall and yellow dwarf coconuts at different times after salt treatment (0, 5, 10, 20, and 30 days). The results revealed continuously elevated levels in MDA (Figure 1C) and soluble sugars (Figure 1D) at all time points, while SOD levels increased until day 10 and then decreased (Figure 1B). In contrast, the concentration of H_2_O_2_ remained at a consistent high level and exhibited no noteworthy alterations at different time points after salt treatment (Figure 1E). Notably, the most profound changes in all physiological characteristics were observed between the 5th and 10th days, underscoring the pivotal roles of these two time points in the response of the two coconut varieties to salt stress.

### 2.2. Transcriptome Sequencing

The spears from four different coconut cultivars were harvested at various time points (0, 1, 5, and 10 days) subsequent to their exposure to saline treatment. Collected samples were then prepared for transcriptome sequencing using Illumina technology, resulting in the generation of data for 48 transcriptomes (Table 1). In total, 278.38 Gb of transcriptome read data was obtained, with an average of 5.8 Gb per transcriptome. A total of 930.44 million clean reads were obtained, with an average of 19.38 million raw reads per transcriptome. The Q30 scores ranged from 92.3% to 94.79%, with a mean value of 94.08%. All the data were deposited in the China National GeneBank (CNGB) under bioproject number CNP0004949.

### 2.3. Gene Expression Pattern after Salt Treatment

Based on transcriptome data, we examined a total of 29,109 annotated genes at least once each, after salt treatment. We compared the transcriptome data of four coconut varieties at four sampling time points of 0, 1, 5, and 10 days, in chronological order and pairwise, to determine their expression trends. The expression trends of these genes are shown in Table 2. In Hainan Tall, aromatic dwarf, red dwarf, and yellow dwarf coconut, 914, 1157, 1068, and 967 genes increased in expression, respectively, at 0, 1, 5, and 10 days subsequent to salt treatment. Six of these genes were shared among all four coconut varieties, including serine decarboxylase 1, starch synthase 3, an uncharacterized gene, nudix hydrolase 8, and glyceraldehyde-3-phosphate dehydrogenase. Conversely, a total of 2112 genes were downregulated in Hainan Tall, 1087 in aromatic dwarf, 2699 in red dwarf, and 2028 in yellow dwarf coconut at 0, 1, 5, and 10 days after salt treatment. Among these, 107 genes were shared among all four coconut varieties.

A total of 746 genes were annotated as being upregulated or downregulated by at least twofold. Multiple genes (listed in Table 1) were found to be involved in various cellular processes, including cell wall biosynthesis (15 genes), fatty acid biosynthesis and metabolism (40 genes), soluble small molecules (3 genes), antioxidant activity (17 genes), transporter function (16 genes), and transcriptional regulation (52 genes). Notably, among the 15 genes involved in cell wall biosynthesis, the xyloglucan endotransglucosylase gene (XTH2, which is related to cell wall loosening) was upregulated more than fivefold following salt treatment, with an RPKM value reaching 276 at 5 days after salt treatment in Hainan Tall coconut. Furthermore, a total of 40 genes involved in fatty acid biosynthesis and metabolism showed at least a twofold increase in expression. Interestingly, 11 genes (27.5%) involved in the biosynthesis of cuticular wax were found, including CYP86A4, HTH, CER1, CER2, CER3, DCR, GPAT4, LTP3, LTP4, and LTP5. In particular, in Hainan Tall coconut, the expression of CER2 was induced over sixfold, with an RPKM value of up to 205, after a 10-day salt treatment.

Based on differential gene expression data among four different coconuts varieties under salt stress, 52 genes related to transcription factors were identified. BHLH transcription factors BHLH1-4 and BHLH6; MYB transcription factors MYB2 and MYB4; ethylene response factors ERF2, ERF4, and ERF6; NAC transcription factor NAC2-8; WRKY transcription factor WRKY1; and zinc finger transcription factor ZAT were induced. However, WRKY transcription factor WRKY2; BHLH transcription factor BHLH5; ethylene response factors ERF1, ERF3, and ERF5; MYB transcription factor MYB1; and transcription factor TT2 were downregulated. In addition, WRKY transcription factor WRKY2 showed a trend of first decreasing and then increasing in expression.

### 2.4. Enrichment Analysis

It was found that differentially expressed genes were enriched in several KEGG pathways, including “metabolic pathways”; “flavonoid biosynthesis”; “biosynthesis of secondary metabolites”; “ribosome”, “photosynthesis-antenna proteins”; “photosynthesis”; “biosynthesis of amino acids”; “cutin, suberine, and wax biosynthesis”; “phenylpropanoid biosynthesis”; and “carbon fixation in photosynthetic organisms”. The KEGG enrichment analysis indicated the participation of cutin, suberine, and wax biosynthesis, a pathway known to be linked with stress resilience in certain species. This suggests the potential association of cutin and wax biosynthesis with salt tolerance in coconut palms.

### 2.5. Cutin, Suberine, and Wax Biosynthesis

Based on the annotated results, we have identified candidate genes that may be involved in cutin biosynthesis, such as the CER and CYP96B5 genes. Protein CYP96B5 catalyzes the conversion of long-chain alkanes to long-chain primary alcohols; then, the long-chain primary alcohols are converted to long-chain ketones, which are epidermal waxy components [43]. We found that the CER1 gene had a consistently low expression level across all four time points following salt treatment in the four coconut varieties (Figure 2). However, we observed an interesting difference in the expression patterns of the CER2 gene across the different coconut varieties. Specifically, in the Hainan Tall and the aromatic varieties, the CER2 gene was induced and upregulated after salt treatment. In the case of Hainan Tall coconut, we observed a continuing increase in CER2 expression levels, with the RPKM value reaching its highest value (205) at 10 days after salt treatment. In the aromatic coconut variety, we observed that the RPKM value of the CER2 gene peaked at 5 days after salt treatment, reaching a maximum of 141.22. However, for yellow dwarf and red dwarf coconut varieties, the expression level of the CER2 gene was low at four different time points after exposure to salt treatment. On the other hand, the CER3 genes demonstrated a similar expression pattern with CER2 amongst all four coconut varieties, with the highest expression level (only 37.4) occurring at 1 day after salt treatment in aromatic coconut. Moreover, the CYP96B5 gene had the highest expression level at 5 days after exposure to salt treatment in both tall and aromatic coconut varieties (RPKM: 44.35 and 84.28, respectively). However, the expression level of CYP96B5 was very low at different time points after salt treatment for the yellow dwarf and red dwarf coconut varieties. In conclusion, the CER (cutin biosynthesis 2) gene may play a crucial role in the biosynthesis of cutin wax and could contribute to the divergence in salt tolerance between tall and dwarf coconut varieties.

Meanwhile, we also investigated the expression of candidate genes involved in wax biosynthesis in response to salt treatment. Our analysis identified six candidate genes involved in cuticle biosynthesis, including the Glycerol-3-phosphate acyltransferase (GPAT), CYP86, HOTHEAD (HTH), DEFECTIVE IN CUTICULAR RIDGES (DCR), and non-specific lipid transfer protein (LTP) genes (Figure 3). Notably, GPAT and LTP genes, which are involved in the first and last steps of wax biosynthesis, respectively, showed consistently high expression levels across all four time points after salt treatment in all four coconut varieties. This suggests that GPAT and LTP genes are constitutively expressed and play a primary and necessary role in coconut cells. In addition, three candidate genes, CYP86, HTH, and DCR, were significantly upregulated at 1 or 5 days after salt treatment in Hainan Tall and aromatic coconut, while they maintained low expression levels at all four stages in red and yellow dwarf coconut.

### 2.6. Cytological Examination of the Cuticle and Epicuticular Layer

Plant cuticle wax covers the surfaces of plant organs, serving as a protective barrier to reduce water loss. Studies have indicated that the production of cuticle wax can elevate in response to environmental stress, resulting in its buildup.

Changes in the cytology of the cuticle and epicuticular layer at four stages following salt treatment in four different coconut varieties were observed. The thickness of the cuticle layer significantly increased (*p* = 0.004) from 81.89 on day 0 to 134.18 on day 10 after salt treatment was applied to Hainan Tall coconut varieties (Figure 4A–C); the thickness increased by 63.85%. However, there was no significant change in the thickness of the cuticle layer at different stages following salt treatment in the yellow dwarf coconut varieties (Figure 4D–F). These results are consistent with differences in gene expression involved in the biosynthesis of the cuticle layer across the two coconut varieties. Under salt treatment, there was an increased capacity for scavenging ROS and a higher abundance of cuticle accumulation in tall coconut.

A scanning electron microscope was used to observe the cuticle wax crystal patterns on the surfaces of leaves of Hainan Tall and yellow dwarf coconuts after salt treatments of varying durations. Prior to treatment, the leaf surfaces of both coconut varieties were densely covered with wax crystals (Figure 4G,I). For the yellow dwarf coconut, the salt treatment for 10 days resulted in sparser wax crystals compared to Hainan Tall. Therefore, it is possible that the 10-day salt treatment may result in severe damage to the cuticular wax in both coconut varieties, but more so in yellow dwarf coconut (Figure 4H,J). The results suggested that salt treatment leads to a reduction in the wax crystals that cover the leaf surface, and the speed of this reduction may be associated with the tolerance to salt damage.

## 3. Discussion

Coconut (*Cocos nucifera*, 2n = 16) is an important tropical crop that exhibits high salt tolerance due to its ancient dissemination style, aided by ocean currents. Tall coconut varieties, which have higher fiber content, were preferred for spreading coconuts across oceans. As a result, the Hainan Tall coconut exhibits higher salt tolerance compared to dwarf coconut, which is supported by our findings. Our study utilized transcriptome data and cytological observations to elucidate the differences in salt tolerance between four coconut varieties.

As next-generation sequencing technology continues to develop, transcriptome sequencing has become an important strategy for studying genome-wide changes in gene expression dynamics. Comparative transcriptome analysis has proven to be an effective approach for elucidating the molecular mechanisms and specific pathways responsible for differences in resistance between different plant varieties. When plants are subjected to salt stress, specific transcription factors in their bodies activate salt-stress-related gene expression. In the MYB transcription factor family, MYB1, MYB2, and MYB4 are significantly induced. MYB1 is involved in the activation of the chalcone synthase gene (CHS) and dihydroflavonol reductase gene (DFR) [44]. CHSs are key enzymes of the phenylpropanoid pathway, and the phenylpropanoid pathway is the starting point for the synthesis of flavonoids and flavonols [45]. DFR plays a key role in the biosynthesis of anthocyanins and proanthocyanidins (belonging to flavonoids) [46]. Flavonoids are phenolic substances isolated from a wide range of vascular plants and can enhance plant oxidative resistance by clearing oxygen free radicals [47]. In the ethylene response transcription factor family, ERF2, ERF4, and ERF6 were induced, while ERF5 was downregulated. Many ERF transcription factors have been confirmed to be important regulatory factors in salt stress response pathways. For example, the overexpression of the JcERF1 [48] gene improves the salt tolerance in transgenic tobacco, and the ERF1 [49] gene in rice can induce a salt stress response by amplifying the MAPK cascade signal. In the NAC transcription factor family, except for NAC1, the other seven NAC transcription factors were induced. Among them, NAC2 [50], NAC3 [51] and NAC6 [52] are involved in plant stress responses under salt stress. In our study, we also found that the expression of NAC and ERF genes was induced, which is consistent with the previous literature. Apart from the common transcription factor gene families related to salt stress, zinc finger transcription factor ZAT and transcription factor TT2 are also worth noting. The expression of zinc finger transcription factor ZAT was induced, while the expression of transcription factor TT2 showed a gradual decline but still maintained a relatively high level of expression. Zinc finger transcription factor ZAT can activate the expression of stress genes (such as the gene lipoxygenase 3(LOX3), related to the biosynthesis of jasmonic acid in Arabidopsis thaliana [53]) under light stress, oxidative stress, and high salt stress [54]. Transcription factor TT2 is involved in the activation of colorless anthocyanin reductase [55]. Colorless anthocyanin reductase can catalyze the formation of catechins ((+)-Afzelechin, (+)-Catechin, and (+)-Gallocatechin) from flavonoids such as leucopelargonidin, leucocyanidin, and leucodelphinidin. Catechins have antioxidant and antibacterial effects and can eliminate oxygen free radicals produced in plants under salt stress, thus protecting plant cells [56].

The thickening of the cuticle layer is a common response to salt stress in plants. The ECERIFERUM (CER) wax synthesis gene can catalyze the conversion of long-chain aldehydes into long-chain alkanes during the wax synthesis process. The expression of cuticle biosynthesis-related genes, including CYP86A4, HTH, CER1, CER2, and CER3, was significantly induced in the four different coconut varieties. Their gene expression levels continued to rise after salt treatment or showed an upward trend within 1 or 5 days, followed by a decrease. In Hainan Tall and aromatic coconuts, this expression trend is obvious. The lipid biosynthesis-related gene CER2 showed a difference in expression of up to 6 times and an RPKM value of over 100. Mutants of the BAHD acyltransferase gene DCR in Arabidopsis displayed typical cuticle defects such as epidermal cell differentiation and a deficiency in 9(10), 16-dihydroxyhexadecanoic acid; in addition, they became more susceptible to salt and osmotic stress. The expression of the BAHD acyltransferase gene DCR was also significantly induced in Hainan Tall and aromatic coconut varieties, while it was not induced in yellow dwarf and red dwarf coconuts. Glycerol-3-phosphate acyltransferase catalyzes the acylation of 3-phosphoglycerol to generate phosphatidic acid, which is the first step in cuticle biosynthesis [57]. Mutants of the Arabidopsis 3-phosphate acyltransferase gene GPAT5 had a lower germination rate under salt stress compared to wild-type, indicating that GPAT is involved in the construction of plant cuticles and responds to salt stress, accordingly [58]. The expression of the four coconut varieties’ 3-phosphate acyltransferase gene GPAT was also significantly induced under salt stress, consistent with the findings of previous research. The synthesis of lipids, facilitated by enzymes like GPAT, plays a critical role in initiating lipid synthesis [59], glycerolipid biosynthesis, oil production, flower development, and stress response [60]. It is a dynamic process that contributes to the structural integrity of cell membranes, the formation of protective layers such as waxes, the storage of energy in the form of oils, and the establishment of lipid reserves for future energy needs [61]. Understanding the regulation of these pathways is essential for developing effective strategies to manipulate lipid metabolism for therapeutic and agricultural purposes. Moreover, non-specific lipid transfer proteins can export large amounts of lipids synthesized inside cells to the cell surface, where they form waxes on the surfaces of cells [62].

## 4. Material and Methods

### 4.1. Plant Materials

Four coconut varieties, specifically Hainan Tall, yellow dwarf, red dwarf, and aromatic coconut, from China, Malaysia, and Thailand, respectively, were cultivated within the same nurseries. In each variety, 12 plants were germinated and grown in the same nursery. Subsequently, seven- to eight-month-old seedlings (plant height about 80 cm and weight about 2–2.5 kg) were selected for salt treatment; these included 12 Hainan Tall, 12 yellow dwarf, 12 red dwarf, and 12 aromatic coconut plants. Prior to treatments, these coconut varieties were placed in a bucket with a drain hole at the bottom, covered with soil (mix of local soil with coconut bran at a 3:1 ratio) and acclimatized in a 26 °C growth chamber for 1 day. As controls for RNA extraction, spear leaf samples were obtained from three separate replicates. The remaining six groups, each comprising three seedling replicates, were exposed to treatment with a 300 mmol/L NaCl solution for 1, 5, and 10 days before sampling. Afterwards, under natural sunlight irradiation, the seedlings were irrigated with 500 mL of saline water every 2 days. Spear leaves were collected from both the control and salt-treated seedlings and immediately frozen in liquid nitrogen.

### 4.2. RNA Extraction

Total RNA was extracted separately from the control and salt-treated leaf samples using the MRIP method as described by Xiao et al. (2012) [63]. Subsequently, equivalent quantities of the extracted RNA were mixed. The preparation of the MRIP extraction buffer involved the combination of the following components to create a 100 mL solution: 3.5 g of ammonium thiocyanate, 9.44 g of guanidine thiocyanate, 3.33 mL of 3 mol/L sodium acetate (pH 5.2), and 38 mL of phenol. The pH of the buffer was subsequently adjusted to 5.0. This procedure adhered to the protocol described by Xiao et al. (2012) [63].

### 4.3. Illumina Sequencing and De Novo Assembly

The initial step involved the fragmentation of purified mRNA, achieved through the application of divalent cations at an elevated temperature. These resulting shorter fragments were employed as templates for the synthesis of the first-strand cDNA, utilizing hexamer primers and superscript III (Invitrogen, Carlsbad, CA, USA). Subsequently, the second-strand cDNA was synthesized in a solution containing buffer, dNTP, RNaseH, and DNA polymerase I. The synthesized cDNA was then purified using a QiaQuick PCR extraction kit (Qiagen, Hilden, Germany) with a buffer containing dNTP, RNaseH, and DNA polymerase I. To facilitate end reparation and poly(A) addition, EB buffer was utilized to resolve these short fragments. Sequence adaptors were ligated to both ends of the short cDNA sequences, and appropriately sized cDNA fragments were selected for PCR amplification based on the results of agarose gel electrophoresis. Finally, the established library was sequenced using an Illumina Hiseq 2000. The paired-end library was prepared following the protocol provided by the Paired-End Sample Preparation Kit (Illumina, San Diego, CA, USA).

### 4.4. Calculating Gene Expression Level

The Bowtie 2 software (Version 2.4.1) was utilized to map the clean reads to the coconut reference genome. Subsequently, the RPKM (reads per kilobase million) value was calculated using the RSEM software (Version 1.3.3) with the following formula:$$\bar{l}(t)=\sum_{i=1}^{l(t)}F(i)(l(t)-i+1)$$

*F* represents fragment and *l(t)* represents length of transcript. A Venn diagram was drawn in the following website (http://bioinformatics.psb.ugent.be/webtools/Venn (accessed on 20 October 2019)).

The log_2_ ratio was utilized to assess the gene expression differences between the salt treatment (1, 5, and 10 days after salt treatment) and control conditions. When log_2_ ≥ 2 or log_2_ ≤ −2, these genes were identified as differentially expressed genes. According to the gene ID of the differentially expressed genes screened, the KEGG and GO function annotations of the differentially expressed genes in coconut were retrieved from GigaScience database (https://academic.oup.com/gigascience (accessed on 13 June 2020)).The KEGG enrichment of differentially expressed genes was analyzed using the KOBAS online tool (http://kobas.cbi.pku.edu.cn/kobas3 (accessed on 10 July 2020)). Finally, the BLAST online tool was used to compare the gene function annotation of the NR library.

### 4.5. Cytological Observation

The leaf margins of Hainan Tall and yellow dwarf coconut varieties were harvested and preserved in Carnoy’s solution for wax examination. These samples underwent a sequence of procedures which included—as integral stages in the processing workflow—dehydration, clearing, wax infiltration, and embedding. Intact sections of these specimens were specifically chosen for microscopic observation. As for scanning electron microscope (SEM) analysis, the leaf margins from both the control and salt-stressed Hainan Tall and yellow dwarf coconut specimens were subjected to a dual-fixation process, followed by dehydration, and ultimately coated with a gold–palladium layer. Again, intact sections were meticulously selected for observation employing SEM.

## Figures and Tables

**Figure 1 plants-13-00141-f001:**
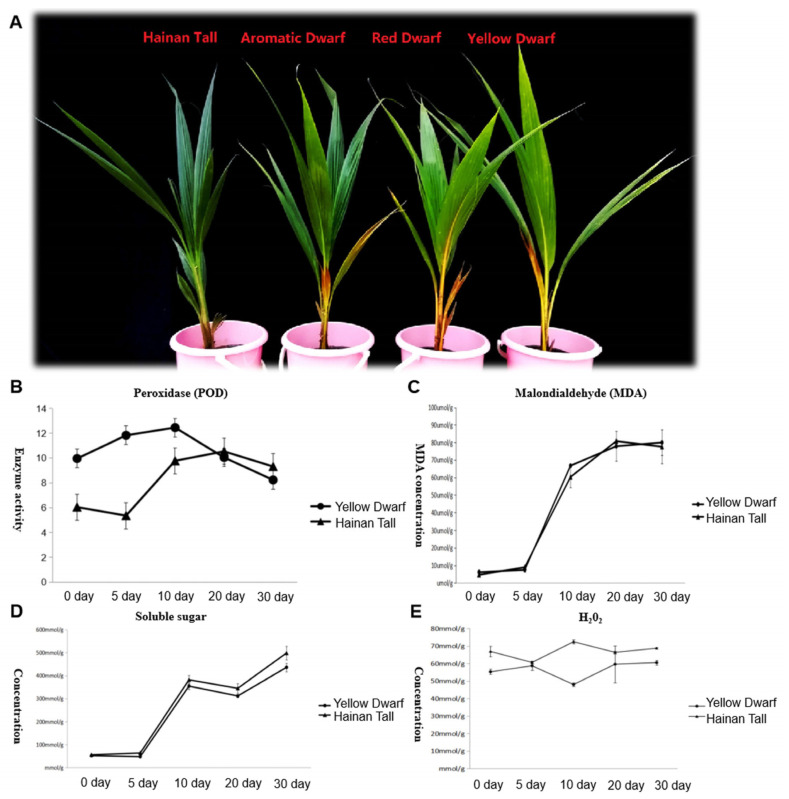
Illustration of the phenotype and physiological changes of four different coconut varieties, including Hainan Tall, aromatic dwarf, red dwarf, and yellow dwarf, at various time points after salt treatment. Here, (**A**) shows the visible phenotype change observed in response to salt stress; (**B**) illustrates the change in SOD activity over time after salt treatment; (**C**) shows the change in MDA concentration at different time points after salt treatment; (**D**) illustrates the change in soluble sugar concentration over time after salt stress; and (**E**) shows the change in H_2_O_2_ concentration at various time points after salt treatment. Data means (±SD) were calculated from three replications and represented by error bars. The LSD (least significant difference) test (*p* < 0.05) on panels (**B**–**E**) showed significant differences between the treatments (*p* < 0.05). The letters above the line in the figure (**B**–**D**) represent yellow dwarf coconut, while the letters below the line represent Hainan Tall coconut. Three biological replicates were set for each sample at each sampling point. The letters above the line in (**E**) represent Hainan Tall coconut and the letters below the line represent yellow dwarf coconut.

**Figure 2 plants-13-00141-f002:**
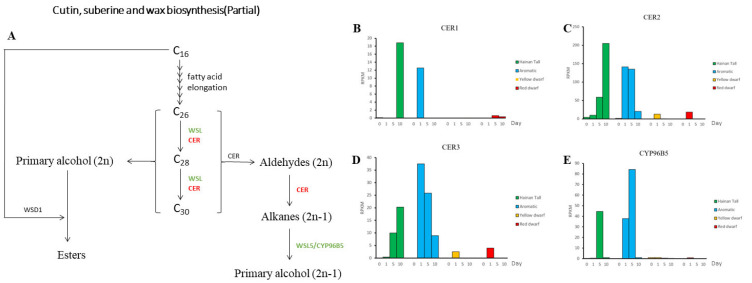
The expression patterns of candidate genes associated with cutin and wax biosynthesis were analyzed across four varieties of coconut, namely Hainan Tall, aromatic coconut, yellow dwarf, and red dwarf. (**A**) Cutin, suberine, and wax biosynthesis pathway. (**B**) Expression of gene CER1 in four coconut varieties. (**C**) Expression of gene CER2 in four coconut varieties. (**D**) Expression of gene CER3 in four coconut varieties. (**E**) Expression of gene CYP96B5 in four coconut varieties.

**Figure 3 plants-13-00141-f003:**
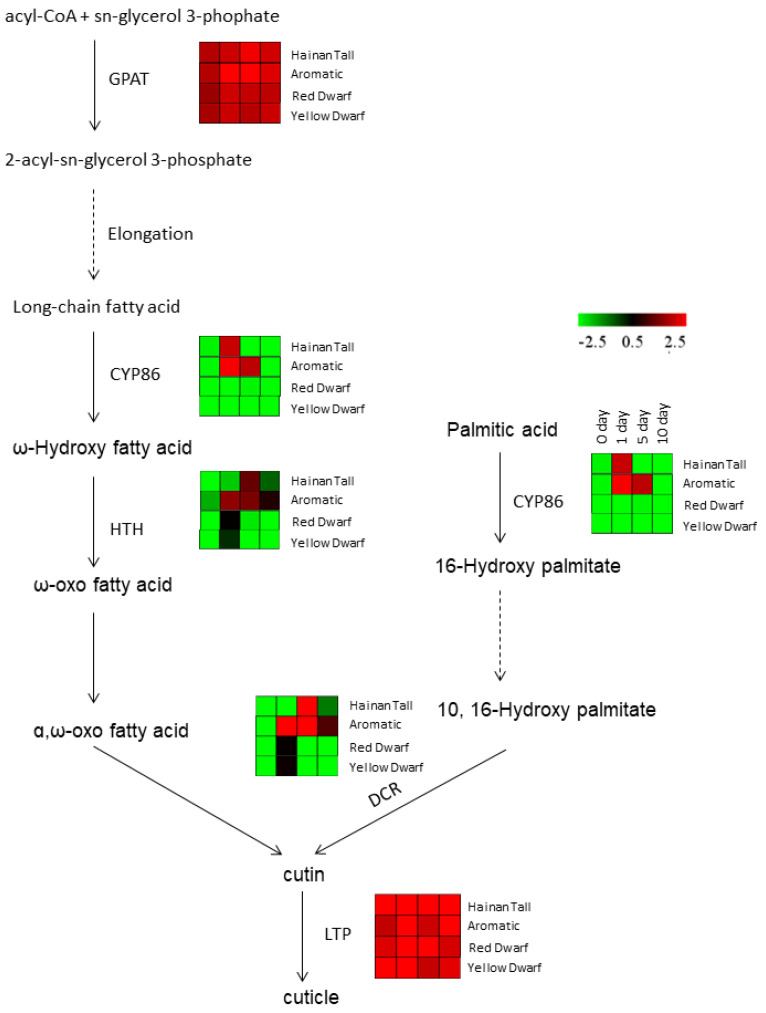
The expression patterns of candidate genes that participate in wax biosynthesis across four coconut varieties: Hainan Tall, aromatic coconut, yellow dwarf, and red dwarf.

**Figure 4 plants-13-00141-f004:**
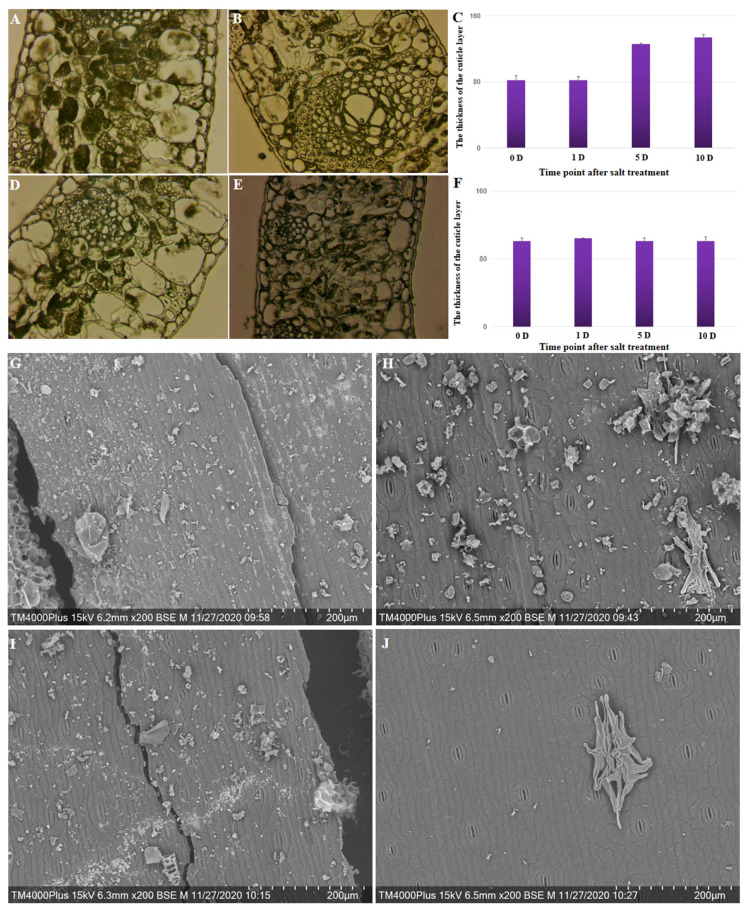
The cytological and scanning electron observations of the circular wax on the leaf edge of Hainan Tall and yellow dwarf coconut under control and salt stress conditions. (**A**) Cytological observation of leaf edge 0 days after salt treatment in Hainan Tall coconut. (**B**) Cytological observation of leaf edge 10 days after salt treatment in Hainan Tall coconut. (**C**) The thickness of the cuticle layer of the leaf edge at 0, 1, 5, and 10 days after salt treatment in Hainan Tall coconut. (**D**) Cytological observation of leaf edge at 0 days after salt treatment in yellow dwarf coconut. (**E**) Cytological observation of leaf edge at 10 days after salt treatment in yellow dwarf coconut. (**F**) The thickness of the cuticle layer of the leaf edge at 0, 1, 5, and 10 days after salt treatment in yellow dwarf coconut. (**G**) Scanning electron observation of leaf edge at 0 days after salt treatment in Hainan Tall coconut. (**H**) Scanning electron observation of leaf edge at 10 days after salt treatment in Hainan Tall coconut. (**I**) Scanning electron observation of leaf edge at 0 days after salt treatment in yellow dwarf coconut. (**J**) Scanning electron observation of leaf edge at 10 days after salt treatment in yellow dwarf coconut.

**Table 1 plants-13-00141-t001:** Transcriptome statistics of four coconut varieties after exposure to salt stress for 0, 1, 5, and 10 days.

		Rep 1			Rep 2			Rep 3	
	Clean Read Number (Million)	Nucleotide Bases (Gb)	Q20 (%)	Clean Read Number (Million)	Nucleotide Bases (Gb)	Q20 (%)	Clean Read Number (Million)	Nucleotide Bases (Gb)	Q20 (%)
Hainan_Tall_0_day	19.83	5.93	97.9	19.18	5.73	98.08	18.53	5.54	97.95
Red_Dwarf_0_day	19.79	5.92	97.86	20.89	6.24	98.06	21.08	6.30	97.95
Yellow_Dwarf_0_day	21.31	6.37	98.14	22.03	6.59	97.89	17.38	5.20	98.12
Aromatic_0_day	20.71	6.19	98.01	18.94	5.67	97.86	17.49	5.23	97.93
Hainan_Tall_1_day	18.01	5.38	97.98	18.98	5.68	97.97	24.85	7.43	97.92
Red_Dwarf_1_day	19.56	5.85	98.03	18.27	5.46	98.16	22.76	6.80	98.04
Yellow_Dwarf_1_day	18.43	5.51	97.91	19.98	5.97	97.77	18.12	5.42	97.98
Aromatic_1_day	17.64	5.27	98.04	22.21	6.64	97.39	21.08	6.30	97.67
Hainan_Tall_5_day	18.08	5.41	97.87	20.15	6.03	97.79	17.26	5.17	97.94
Red_Dwarf_5_day	19.52	5.85	97.74	21.06	6.31	97.81	21.20	6.34	97.72
Yellow_Dwarf_5_day	20.34	6.09	97.81	16.99	5.09	97.66	18.66	5.59	97.85
Aromatic_5_day	19.37	5.80	97.78	18.76	5.62	97.83	17.88	5.36	97.87
Hainan_Tall_10_day	21.22	6.35	97.83	18.11	5.43	97.93	19.49	5.84	97.75
Red_Dwarf_10_day	18.16	5.44	97.81	18.47	5.53	97.6	17.73	5.31	97.82
Yellow_Dwarf_10_day	18.39	5.51	97.81	18.62	5.58	97	17.92	5.37	97.66
Aromatic_10_day	18.86	5.65	97.71	18.88	5.66	97.69	18.28	5.47	97.79

**Table 2 plants-13-00141-t002:** The expression patterns of annotated genes at different time points after salt treatment.

Trend	Trend Fig.	Variety	Expression Number	Shared Gene Number	Venn Diagram
UUU	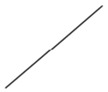	Hainan Tall	914	6	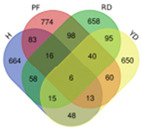
Aromatic	1157
Red dwarf	1068
Yellow dwarf	967
UUD	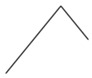	Hainan Tall	2808	22	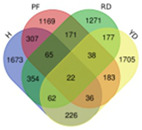
Aromatic	2091
Red dwarf	2302
Yellow dwarf	2568
UDD	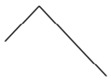	Hainan Tall	2431	102	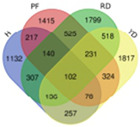
Aromatic	3148
Red dwarf	3929
Yellow dwarf	3606
UDU	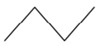	Hainan Tall	3991	57	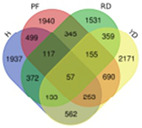
Aromatic	4232
Red dwarf	3221
Yellow dwarf	4543
DDD	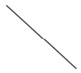	Hainan Tall	2112	107	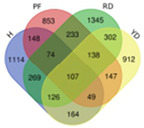
Aromatic	1087
Red dwarf	2699
Yellow dwarf	2028
DDU	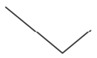	Hainan Tall	4770		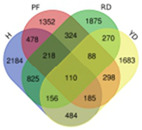
Aromatic	3189	
Red dwarf	4003	110
Yellow dwarf	3403	
DUD	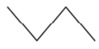	Hainan Tall	4471	62	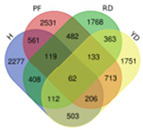
Aromatic	5072
Red dwarf	3584
Yellow dwarf	4096
DUU	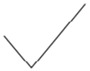	Hainan Tall	2586		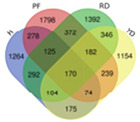
Aromatic	3383	
Red dwarf	3118	170
Yellow dwarf	2576	

Note: In the “Trend” column, the first letter (U: up, D: down) represents the trend in transcriptional expression change from day 1 to day 0, the second letter represents the trend in transcriptional expression change from day 5 to day 1, and the third letter represents the trend in transcriptional expression change from day 10 to day 5.

## Data Availability

Transcriptome short reads were deposited in the China National GeneBank (CNGB); the bioproject number is CNP0004949.

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
