# Peer review of "Comparative Transcriptome Analysis Provides Insights into the Effect of Epicuticular Wax Accumulation on Salt Stress in Coconuts"

_plants, 2024, doi:10.3390/plants13010141_

Round 1
Reviewer 1 Report
Comments and Suggestions for Authors
This works presents interesting research on the genes that are regulated for salt-stress tolerance in coconut varieties, and it is complemented with some physiological responses observed in the treated plants. To improve the quality of the manuscript please consider the following observations:
Introduction:
- Line 36, please provide examples of the different types of salinity stress.
- Line 38, What geographical sites present chloride deficiency?
- Line 45-46, Have the number of salt-affected soils increased in the recent years? What are the causes?
- Paragraphs from line 54 to 71. Please provide more information regarding the genes and the enzyme coded by those genes, and the metabolites that are produced by those enzyme pathways.
- Line 57, “Suaeda maritima” should be in italics.
- Line 74, correct the word “gactor”, it should be “factor”.
Results:
- Line 81, how you determinate that 300 mmol/L of NaCl was the most suitable for the experiments? Is that the NaCl concentration found in the lands near the sea? Or subjected to sea spray/breeze?
- Paragraph of line 80-95. The labels referencing Figure 2C and Figure 2D are wrong.
- In this paragraph are mentioned the four varieties of coconut assayed. But it would be more appropriate to describe the varieties in the introduction, their characteristics, their genetic relationship, agricultural value and productivity, this is to justify the use of those plants varieties.
- Figure 1 (B to E) has the graphics only for Hainan Tall and Aromatic Dwarf, but is lacking the results for the other dwarf varieties. Please include that information in the figures or in the text.
- Section 2.3, about the gene expression it would have been interesting to look for other genes related to metabolic processed such as sugar metabolism (cytoplasmic and mitochondrial) to determine the carbon redirection into lipid biosynthesis. Also, it interesting to see what is happening with enzymes in the reticulum and the transport of vesicles.
- Line 164, please describe the enzyme coded by CYP96B5.
- Figure 3, on the patway of the right side it says “Palmitic aicd”, please correct the typo.
- Section 2.6, line 212, there is mentioned the increase in thickness of cuticle. Please include the fld of change or % of increase that those values represent. Also clearly indicate what coconut variety showed that change.
- Paragraph from 229 to 231. Please move this paragraph to the beginning of this section.
- Figure 4 (Page 10). There is not panel D. And in the figure legend there in not explanation for panel E. Please verify and correct the labeling and description of the photographs.
Discussion:
- The section should be titled “Discussion”, not “Discussions”.
- Line 262, please add more information regarding the role of chalcone synthase and the synthesis of antioxidant molecules and tolerance compounds.
- Line 263, please further explain the role of ethylene for the activation of stress tolerance-related mechanisms.
- Line 273, please include examples of stress-related genes regulated by ZAT.
- Paragraph from line 280. There are several genes related to was synthesis. Is the pathway completely upregulated? What genes were not upregulated? What kind of components for the wax and cuticle are product of those genes?
- At the final paragraph of the discussion please include more information about the synthesis of lipids after the explanation of the enzyme GPAT, in this context is interesting to correlate the lipids redirection into several processes such as membrane repair, wax biosynthesis, oil biosynthesis, and lipid storage.
Materials and methods:
- In the plant material section please include what age (days or weeks) were the plants specimens, and their characteristics such as lengths and weight.
- Include the kind of substrate, soil or culture medium for the plants culture. The water regime and the volume and frequency of NaCl application, the light photoperiod, light source (natural sunlight, LED, fluorescent, halogen).
Comments on the Quality of English LanguageThere are minor errors.
Author Response
Reviewer 1
Introduction:
- - Line 36, please provide examples of the different types of salinity stress.
>>>Responses: Citation inserted for this sentence: Gong S., Chen S. (2018) The Change of Osmoregulation Substance Content in Different Varieties of Coconut at the Nursery Stage under Salt Stress. Chinese Journal of Tropical Agriculture, 38(06):1-5. This is a Chinese reference, we have translated it into English, this one can be found as reference.
- - Line 38, What geographical sites present chloride deficiency?
>>>Responses: Line 38. The location 'Davao Mindanao' in the Philippines has been added in the text.This one can be found as reference [12]
- - Line 45-46, Have the number of salt-affected soils increased in the recent years? What are the causes?
>>>Responses: Line 49. The annual increase in the quantity of saline-alkali land and the causes thereof have been added. This one can be found in reference [18]
- - Paragraphs from line 54 to 71. Please provide more information regarding the genes and the enzyme coded by those genes, and the metabolites that are produced by those enzyme pathways.
>>>Responses: Additional information regarding other enzyme genes have been added into line 59-68 of revised manuscript.
- - Line 57, “Suaeda maritima” should be in italics.
>>>Responses: Thank you for your suggestions, changed at line 62.
- - Line 74, correct the word “gactor”, it should be “factor”.
>>>Responses: Thank you for your suggestions, changed at line 76.
Results:
- - Line 81, how you determinate that 300 mmol/L of NaCl was the most suitable for the experiments? Is that the NaCl concentration found in the lands near the sea? Or subjected to sea spray/breeze?
>>>Responses: We conducted preliminary experiments with gradient concentrations of NaCl, namely 300mmol/L, 400mmol/L, 600mmol/L, and 800mmol/L. A 300 mmol/L sodium chloride treatment was found to be the optimal salinity level for all four coconut varieties within an acceptable range. Therefore, we used this concentration in the experiment.
- - Paragraph of line 80-95. The labels referencing Figure 2C and Figure 2D are wrong.
>>>Responses: Line 86-89. In the main text, Figure 2C and Figure 2D have been corrected to the appropriate figures, namely Figure 1C and Figure 1D. same to Figure 2B and Figure 2E, they have also been amended to Figure 1B and Figure 1E.
- - In this paragraph are mentioned the four varieties of coconut assayed. But it would be more appropriate to describe the varieties in the introduction, their characteristics, their genetic relationship, agricultural value and productivity, this is to justify the use of those plants varieties.
>>>Responses: Line 42-46. Thank you for your suggestion, Details regarding the description of yellow dwarfs, red dwarfs, and aromatic coconuts have been moved into the “Introduction” part of revised manuscript.
- - Figure 1 (B to E) has the graphics only for Hainan Tall and Aromatic Dwarf, but is lacking the results for the other dwarf varieties. Please include that information in the figures or in the text.
>>>Responses: I'm sorry that we can only work on the Hainan Tall and Aromatic Dwarf varieties. Therefore, we can only provide information available in the current version. If it was allow in the future, we will continue this research with other dwarf varieties.
- - Section 2.3, about the gene expression it would have been interesting to look for other genes related to metabolic processed such as sugar metabolism (cytoplasmic and mitochondrial) to determine the carbon redirection into lipid biosynthesis. Also, it interesting to see what is happening with enzymes in the reticulum and the transport of vesicles.
>>>Responses: We identified a total of 96 genes involved in Glycolysis, 41 genes involved in lipid biosynthesis and 21 genes in the reticulum and the transport of vesicles in coconut. There are not significant difference between salt treatment and control.
- - Line 164, please describe the enzyme coded by CYP96B5.
>>>Responses: Line 154-165. The explanation of CYP96B5 has been added, detail can be found at section 2.5, reference [40].
13- Figure 3, on the patway of the right side it says “Palmitic aicd”, please correct the typo.
>>>Responses: Figure 3. Thank you for your suggesions. The corresponding revision had been done.
- - Section 2.6, line 212, there is mentioned the increase in thickness of cuticle. Please include the fld of change or % of increase that those values represent. Also clearly indicate what coconut variety showed that change.
>>>Responses: Line197. A growth percentage has been added into the text, indicating a 63.85% increase in the thickness of the stratum corneum. It is noted that this change is attributed to the Hainan high coconut. Refer to Section 2.6, second paragraph for further details
- - Paragraph from 229 to 231. Please move this paragraph to the beginning of this section.
>>>Responses: Line 193. The last paragraph of Section 2.6 has been moved to the beginning of Section 2.6
- - Figure 4 (Page 10). There is not panel D. And in the figure legend there in not explanation for panel E. Please verify and correct the labeling and description of the photographs.
>>>Responses: Line 204, 207, 216-224. The order of images in Figure 4 has been corrected to ABCDEFGHIJ, and the corresponding captions have been revised accordingly
Discussion:
- - The section should be titled “Discussion”, not “Discussions”.
>>>Responses: Line 226. Thank you for your suggestion. The corresponding revision had been done.
- - Line 262, please add more information regarding the role of chalcone synthase and the synthesis of antioxidant molecules and tolerance compounds.
>>>Responses: Line 237-241. Descriptions of the functions of the CHS and DFR genes, as well as the role of flavonoids, have been added. Please refer to the second paragraph of the discussion, citing references 42, 43, and 44.
- - Line 263, please further explain the role of ethylene for the activation of stress tolerance-related mechanisms.
>>>Responses: Line 242-244. Cases of ERF transcription factor response under salt stress have been added, refer to citations 45 and 46 for details.
- - Line 273, please include examples of stress-related genes regulated by ZAT.
>>>Responses: Line 251-252. Related examples have been added, please check [51]
- - Paragraph from line 280. There are several genes related to was synthesis. Is the pathway completely upregulated? What genes were not upregulated? What kind of components for the wax and cuticle are product of those genes?
>>>Response: Line260-261. Detailed explanations have been added, elucidating the expression changes of these genes in four coconut varieties. The roles of these genes in the synthesis of cuticular wax have been previously discussed in the text
- - At the final paragraph of the discussion please include more information about the synthesis of lipids after the explanation of the enzyme GPAT, in this context is interesting to correlate the lipids redirection into several processes such as membrane repair, wax biosynthesis, oil biosynthesis, and lipid storage.
>>>Responses: Thank you for your suggestion, we had added some discussion about the enzyme GPAT in the “discussion” part of revised manuscript.
Materials and methods:
- - In the plant material section please include what age (days or weeks) were the plants specimens, and their characteristics such as lengths and weight.
>>>Responses: Line 276-281. Information on plant height, age in months, and weight has been added to Method 4.1. Please refer to the first modification in Section 4.1 for details
- - Include the kind of substrate, soil or culture medium for the plants culture. The water regime and the volume and frequency of NaCl application, the light photoperiod, light source (natural sunlight, LED, fluorescent, halogen).
>>>Responses: Line 284-285. The application volume and frequency of NaCl have been included. The light source has been updated to natural light. The soil coverage description involves a local soil and coconut coir mixture in a 3:1 ratio, with details provided in the modifications after Section 4.1
Reviewer 2 Report
Comments and Suggestions for Authors
Sun et al´s using transcriptome analysis to identify candidate genes that may be involved in the salt stress in plants, which they indirectly allude to genes that may be involved with the accumulation of the epicuticular waxes.
The results are very poorly analysed, and described. RNA-seq data, though is good data but lacks in depth analysis. Some of my critical comments are:
-
Authors need to clarify that what they mean by different expressed genes or differentially expressed genes.
-
Line 92-94: authors claim that they found profound changes between 5-10th days, but looking at the results in Figure 1, looks like changes from 20-30 days are also important. Can authors explain why they did not included those days for further analysis
-
Related to point #2, also RNA-seq from one of those two days would have been useful as an outlier.
-
Figure 1: Legend needs to be described in detail. How many replicates were used, what does error bar signify? Also no scale bar on figure 1A.
-
Line 140- Authors repeatedly mentioned six fold but with respect to FPKM. I am totally confused as fold changes are usually mentioned only after proper Differential gene expression expression analysis which the study fails to do. Use of FPKM and fold change do not make sense at all.
-
Table 2: The legend is incomplete. It took me a while to understand what does U and D in trend show. And how do you reach those trends if no proper analysis was done.
-
Line 179- Authors refer RPKM. They should be consistent in using either FPKM or RPKM.
-
Line 182-184: Authors conclude that CER2 may play an important role in biosynthesis of cutin wax. This is not supported by evidence that they provided. There was no functional study conducted.
-
Figure 2: Legend should be self explanatory. What is on the left and what is on the right needs to be clarified. Also, I think instead of heatmap, authors should provide bar plot since its only 4 genes whole expression is shown.
-
Quantification and hence interpretation of wax and cuticular layer based on images is not always authentic. Extraction and then quantification of wax would have been useful.
-
Figure 4: Legend is not clear. What are Panels E, G, H. Also i am not clear about whats the difference between panel C and G.
England language needs to be significantly improved.
Author Response
- Authors need to clarify that what they mean by different expressed genes or differentially expressed genes.
>>>Response: Line 308-314. Thank you for your suggestion. Indeed, there is indeed clear definition for differentially expressed genes. In fact, the log2 ratio was applied to compute the gene expression differences between the salt treatment (1, 5, 10 day after salt treatment) and control conditions. When log2 ratio ≥ 2 or log2 ratio ≤ -2, these genes were identified as differentially expressed genes and selected for subsequently KEGG and GO annotation. The corresponding revisions had been done in Line - of “Materials and methods” of revised manuscript.
- Line 92-94: authors claim that they found profound changes between 5-10th days, but looking at the results in Figure 1, looks like changes from 20-30 days are also important. Can authors explain why they did not included those days for further analysis
>>>Responses: It is very regret that we could not collect transcriptome data after 20 days and 30 days after salt treatment. In the study, we had sampled the spear leaf from 0 to 30 days after salt treatment. However, RNA from some samples were qualified for RNA-seq. For coconut, RNA extraction is still difficult because of thick cuticle and abundant inclusions. Obviously, more candidate genes involved in other signal and metabolic pathway may be identified from 20-30 days after salt treatment. However, our data can support some evidences for the effect of epicuticular wax accumulation on salt stress in coconut.
- Related to point #2, also RNA-seq from one of those two days would have been useful as an outlier.
>>>Response: It is very regret that we could not collect transcriptome data after 20 days and 30 days after salt treatment.
- Figure 1: Legend needs to be described in detail. How many replicates were used, what does error bar signify? Also no scale bar on figure 1A.
>>>Responses: Line 101-102. Thank you for your suggestions, the error bar refer to standard deviation. The corresponding revision had been done in Figure 1 legends of revised manuscript. Meanwhile, scale bar had been added into the Figure 1.
- Line 140- Authors repeatedly mentioned six fold but with respect to FPKM. I am totally confused as fold changes are usually mentioned only after proper differential gene expression analysis which the study fails to do. Use of FPKM and fold change do not make sense at all.
>>>Responses: Thank you for your suggestion; in previous manuscript, we had not described the differential expression analysis which we performed. In fact, log2 ratio applied to compute the gene expression differences between the salt treatment (1, 5, 10 day after salt treatment) and control conditions. The corresponding revisions had been done in Line - of “Materials and methods” of revised manuscript. The differential expression of CER2 is log2 ratio (206/4), which is approximately equal to 6. In revised manuscript, the corresponding revision had been done in line and line of revised manuscript.
- Table 2: The legend is imcomplete. It took me a while to understand what does U and D in trend show. And how do you reach those trends if no proper analysis was done.
>>>Responses: Line 142-144. Thank you for your suggestion. Indeedly, the legend is imcomplete. The letter “U” represents up-regulated expression and while “D” represented down-regulated expression. The corresponding revisions had been done in Table 2 legend of revised manuscript.
- Line 179- Authors refer RPKM. They should be consistent in using either FPKM or RPKM
>>>Responses: Thank you for mentioning this point, in revised manuscript, all FPKM have modified into RPKM.
- Line 182 -184: Authors conclude that CER2 may play an important role in biosynthesis of cutin wax. This is not supported by evidence that they provided. There was no functional study conducted.
>>>Responses: Thank you for your suggestion, indeedly, our data can not elucidate that CER2 may play an important role in biosynthesis of cutin wax. But, a higher expression level of genes involved in cutin biosynthesis were detected in tall coconuts after salt treatment. This indicates that the accumulation of cutin wax after salt treatment may contribute to higher salt tolerance in tall coconut compare to dwarf coconut. The corresponding revisions had been done in line of revised manuscript.
- Figure 2: legend should be self explanatory. What in on the left and what is on the right needs to be clarified. Also, I think instead of heatmap, authors should provide bar plot since its only 4 genes whole expression.
>>>Responses: Thank you for your suggestions, Having clarified. Figure 2 had been modified into bar charts.
- Quantification and hence interpretation of wax and cuticular layer based on images is not always authentic. Extraction and then quantification wax would have been useful.
>>>Responses: Thank you for your suggestions, extraction and quantification of wax is useful. However, we won't be able to finish these tasks within the the review period. In the subsequent experiments, we will carry out the corresponding work.
- Figure 4 is not clear. What are panel E, G, H. Also i am not clear about whats the difference between panel C and G.
>>>Responses: Thank you for mentioning this errors. Figure 4 had been revised and correct cited in the text.
Round 2
Reviewer 1 Report
Comments and Suggestions for Authors
The new version of the manuscript includes many improvements in the introduction, results and discussion sections.
All the suggestions were considered and new information was included in the text. The quality of the figures in the results sections has imporved.
Author Response
Dear editor and reviewers
Thank you very much for the critical comments and suggestions from you on our manuscript entitled “Comparative transcriptome analysis provides insights into the effect of epicuticular wax accumulation on salt stress in coconuts” (plants-2738793). We have made thorough editing to the language of manuscript. Meanwhile, we have replace some cites with new references.
Kind regards,
Yong Xiao
Reviewer 2 Report
Comments and Suggestions for Authors
The authors have addressed all my concern. Thank you.
Comments on the Quality of English LanguageMinor English correction still required.
Author Response

(The authors gave the same response as above.)
